# Negative Selection in *Oreochromis niloticus* × *O. aureus* Hybrids Indicates Incompatible Oxidative Phosphorylation (OXPHOS) Proteins

**DOI:** 10.3390/ijms26052089

**Published:** 2025-02-27

**Authors:** Andrey Shirak, Arie Yehuda Curzon, Eyal Seroussi, Moran Gershoni

**Affiliations:** 1Institute of Animal Science, Agricultural Research Organization, Rishon LeTsiyon 75288, Israel; shiraka@volcani.agri.gov.il (A.S.); arie.curzon@mail.huji.ac.il (A.Y.C.); 2Robert H. Smith Faculty of Agriculture, Food and Environment, Hebrew University of Jerusalem, Rehovot 76100, Israel

**Keywords:** hybrid selection, cichlid fish, introgression, mitonuclear incompatibility

## Abstract

Crossing *Oreochromis niloticus* (*On*) females with *O. aureus* (*Oa*) males results in all-male progeny that are essential for effective tilapia aquaculture. However, a reproductive barrier between these species prevents commercial-scale yield. To achieve all-male progeny, the currently used practice is crossing admixed stocks and feeding fry with synthetic androgens. Hybrid tilapias escaping to the wild might impact natural populations. Hybrids competing with wild populations undergo selection for different stressors, e.g., oxygen levels, salinity, and low-temperature tolerance. Forming mitochondrial oxidative phosphorylation (OXPHOS) complexes, mitochondrial (mtDNA) and nuclear DNA (nDNA)-encoded proteins control energy production. Crossbred tilapia have been recorded over 60 years, providing an excellent model for assessing incompatibility between OXPHOS proteins, which are critical for the adaptation of these hybrids. Here, by comparing nonconserved amino acid substitutions, across 116 OXPHOS proteins, between *On* and *Oa*, we developed a panel of 13 species-specific probes. Screening 162 SRA experiments, we noted that 39.5% had a hybrid origin with mtDNA-nDNA allele mismatches. Observing that the frequency of interspecific mtDNA-nDNA allele combinations was significantly (*p* < 10^−4^) lower than expected for three factors, *UQCRC2*, *ATP5C1*, and *COX4B*, we concluded that these findings likely indicated negative selection, cytonuclear incompatibility, and a reproductive barrier.

## 1. Introduction

A notable difference in mutation rates between mitochondrial DNA (mtDNA) and nuclear DNA (nDNA) drives tight coadaptation through natural selection between these genomes [1] while promoting the creation of an interspecific barrier [2]. Hybridization between closely related species might disrupt this adaptation and significantly affect different cellular functions and organismal fitness [3,4,5]. This is evident in genes encoding the mitochondrial oxidative phosphorylation (OXPHOS) system, which is essential for cellular energy production. The OXPHOS system is also a major source of reactive oxygen species (ROS), for which elevated formation leads to molecular damage and oxidative distress, which play important roles in cell signaling [6]. Five OXPHOS protein complexes (denoted by Roman numerals I–V) are generally encoded by nuclear and mitochondrial genomes, excluding complex II, which is solely encoded by nDNA. Proper interactions of these complex subunits are required for mitochondrial electron transport and energy production. In animals, the OXPHOS system is formed by 13 mtDNA-encoded subunits and more than 73 nDNA-encoded subunits [1,7]. The incompatibility between OXPHOS proteins in hybrids could lead to decreased fitness in a wide range of essential functions, such as growth rate [8], reproduction [9] and lifespan [10], all of which induce selective pressures. According to the Dobzhansky–Muller model [11], hybrid incompatibility manifests as negative selection, and cytonuclear incompatibility is a specific type of Dobzhansky–Muller genetic incompatibility caused by improper interactions between mitochondrial and nuclear genomes. Moreover, cytonuclear incompatibility can have strong lethal effects in early development [12]. KR/KC and KA/KS are two ratios that are used to quantify selection pressure on a protein-coding sequence, whereas KR is the radical (nonconserved) amino acid replacement rate; KC is the conserved replacement rate; KA is the nonsynonymous substitution rate; and KS is the synonymous substitution rate [13,14,15]. Early studies have shown that substitutions of amino acids whose physicochemical properties are highly different are less frequent than those of amino acids whose physicochemical properties are more similar [16]. A gene KA/KS ratio higher than one suggests a functional substitution and, therefore, positive selection and adaptation processes. The KR/KC ratio more accurately estimates the selection pressure on distantly related protein-coding sequences [17]. A better correlation with vital functions for KR/KC than for KA/KS has been demonstrated in several studies [18,19,20,21,22]. The KR/KC and KA/KS ratios are correlated, but the degree of correlation depends on the various systems of amino acid grouping on the basis of their similarity [23,24]. Although there are many ways to classify amino acids, they are often sorted into six main classes on the basis of their structure and the general chemical characteristics of their side chains: aliphatic (G, A, V, L, I); hydroxyl or sulfur/selenium-containing (S, C, U, T, M); cyclic (P); aromatic (F, Y, W); basic (H, K, R); and acidic and amide-containing (D, E, N, Q) [25,26]. Even a single nonconserved amino acid substitution in an OXPHOS nDNA-encoded subunit following mating of different populations can lead to incompatibility and significantly affect hybrid offspring, as demonstrated for *Drosophila subobscura* [27].

The genus *Oreochromis* comprises over 40 species. *O. niloticus* (*On*) and *O. aureus* (*Oa*) play important roles in the aquaculture industry, and more than 80% of the world production of farmed tilapia is based on hybrids of these two species [28]. *Oreochromis* cichlid fishes are known as evolutionarily young species with rapid radiation and specialization [29,30]. A comparison of *On* and *Oa* genomic sequences revealed 95–98% and 92.8% nDNA and mtDNA identity, respectively [31,32]. However, different studies have demonstrated significant physiological differences between these species, including differences in oxygen consumption, feeding behavior, and tolerance to salinity and low temperature [33,34,35]. Coping with stress elevates energy and oxygen consumption and it is characterized by extensive protein metabolism and ammonia excretion [36]. Thus, some investigations have demonstrated that the oxygen-to-nitrogen ratio in the water environment critically affects tilapia species such as *Oa*, which inhabit deeper layers of water with lower oxygen concentrations [37]. Both young and adult *Oa* are herbivorous, whereas young *On* are herbivorous, and adults tend to be omnivorous. Significant differences between these species were also observed in terms of tolerance to high and low water temperatures. The minimal semilethal water temperature of *Oa* is 7–9 °C, and that of *On* is at least two degrees higher [38].

The *Oreochromis* species *On*, *Oa*, *O. mossambicus*, and *O. urolepis hornorum* are frequently hybridized for aquaculture purposes. The first experiments and observations of tilapia hybridization were reported in the 1960s [39,40,41]. When reared in commercial ponds, the F1 generations of *Oreochromis* hybrids have strong advantages over parental species, such as fast growth, proficient reproduction, and better adaptation to a wide range of temperatures and salinities [42]. However, excessive use of hybrids promotes their escape to the wild and admixture with natural tilapia populations [43]. One explanation given for the shrinking populations of tilapia in the natural habitat is the cytonuclear incompatibility in the emerging hybrid generations [44].

Admixed populations have often been wrongly referenced as purebred species. For *On* and *Oa*, proofs of such errors can be found in GenBank by analyzing DNA- and RNA-seq data obtained from the SRA database [45]. Therefore, in the present meta-analysis study, we first analyzed SRA data to evaluate the hybridization level by identifying and comparing the OXPHOS coding genes in the closely related species *On* and *Oa* via their mitochondrial and nuclear genome sequences. Second, by identifying nonconserved substitutions between OXPHOS gene-orthologs of these species, we demonstrated negative selection for these substitutions indicating involvement in cytonuclear incompatibility and likely in the reproductive barrier.

## 2. Results

### 2.1. Identifying the OXPHOS Genes in the On and Oa Mitochondrial Genomes

Comparing submissions of the mtDNA-encoded *COX1* gene for *On* and *Oa* with those previously reported [46], we detected high levels (20–40%) of misidentification of these species in the NCBI and BOLD databases. Through analysis of 14 complete mitochondrial genome sequences reported for *On* and *Oa* in the NCBI, we detected five cases of misidentifications, all referred to as *On* but with *Oa* or *O. mossambicus COX1* sequences. The remaining nine mitochondria were used to compare the 13 sequences of mtDNA-encoded OXPHOS proteins between *On* and *Oa* (Appendix A).

### 2.2. Identifying On and Oa nDNA-Encoded OXPHOS Genes in the On and Oa Genomes

*On* and *Oa* genomes in NCBI have annotations for 93 nuclear- and 13 mtDNA-encoded OXPHOS genes reported (Appendix A). As previously reported in other fish genomes, some OXPHOS nDNA-encoded genes are duplicated or triplicated [47]. Thus, their number exceeds the number of proteins usually observed in the mammalian OXPHOS system (<85) [48]. In both tilapia genomes, six genes (*SDHB*, *UQCRFS1*, *COX5A*, *ATP5A1*, *ATP5H*, and *ATP5B*) were duplicated, and three other genes (*NDUFA4*, *COX5B*, and *COX7A2*) were triplicated. For the *On* genome build in GenBank (nucleotide accession GCF_001858045.2), three genes were incorrectly predicted by the computerized tool. These included *NDUFA11* (*LOC100706132*, *solute carrier family 5 member 9*), which merged two genes in contrast to the better annotation of *On* in Ensembl *ENSONIG00000036286 NADH: ubiquinone oxidoreductase subunit A11* gene; *COX6A* (*LOC109204496*, uncharacterized ncRNA), which is annotated in Ensembl as the *ENSOABG00000015352 cytochrome c oxidase subunit 6A1* gene; and *ATP5I*, which was not annotated in GenBank *On* genome, although its mRNA sequence is available as an EST (nucleotide accession GR661800). *ATP5I* is partially annotated as the *ENSONIG00000033968* gene, which has proton–transmembrane transporter activity.

### 2.3. Orthologous Genome Positions of Nuclear Genes

The chromosomal mapping positions of all the nDNA-encoded OXPHOS genes of the *Oa* genome build (Nucleotide accession GCF_013358895.1) are given, excluding *UQCRFS1* and *ATP5G2*, which are mapped to unplaced scaffolds. This arises from incomplete assembly in the terminal regions of LG1 and LG4, respectively. Mapped onto the same LGs in the *On* and *Oa* genomes, the 93 nuclear coding OXPHOS orthologs were scattered over 20 of the 23 LGs of tilapia, with LGs 8, 9 and 21 excluded. However, genes that mapped to LGs 2–5, 7, 12–14, 19, and 22 had substantially different positions because of an inverted base numbering in these orthologous LG builds.

*UQCRFS1* and *ATP5G2* are mapped to unplaced scaffolds number 60 (NW_024108958.1) and 80 (NW_024108978.1), respectively. By searching for the orthology of *On*, we mapped these scaffolds onto LG1 and LG4, respectively. The unplaced *Oa* 311 kb scaffold 60 in addition to *UQCRFS1* includes six well-annotated genes, *CELF1*, *MTFMT*, *RASL12*, *KBTBD13*, *UBAP1LB* and *PDCD7*, which are mapped near *On UQCRFS1* on LG1. Unplaced *Oa* 171 kb scaffold 80 in addition to *ATP5G2* includes uncharacterized *LOC120437321* for ncRNA. *Oa LOC120437321* (XR_005611016.1) was used as the query for a BLASTN search against the *On* genome, which detected a highly similar gene (99%) on LG4. Our comparison of the nuclear OXPHOS genes in the *On* and *Oa* genome builds detected identity in the gene number and orthologous map positions for the 93 identified factors.

### 2.4. OXPHOS Gene Variation

By aligning the protein sequences of *On* and *Oa* 93 nDNA- and 13 mtDNA-encoded OXPHOS factors, we detected 58 and 70 conserved and nonconserved amino acid substitutions, respectively. Although nuclear factors account for 87.7% of all OXPHOS factors, only 24 (41.4%) conserved and 26 (37.1%) nonconserved substitutions were detected in these proteins. Consequently, only 16 of 93 (17.2%) nDNA-encoded factors carried nonconserved substitutions, whereas 76.9% (10/13) of the mtDNA-encoded genes had such substitutions (Table 1). When nDNA- and mtDNA-encoded genes were compared into two distinct groups in which all the members were considered the same entity, KR/KC ratios exceeding 1 (26/24 = 1.08 and 44/34 = 1.29, respectively, suggesting adaptive selection) were detected for both groups (Appendix A). After accounting for gene sequence length, nDNA-encoded OXPHOS proteins presented a conserved amino acid substitution rate that was 7.00-fold lower than that of the mtDNA-encoded OXPHOS proteins, whereas an 8.36-fold reduction was observed for nonconserved substitutions (Appendix A). Thus, the nDNA-encoded OXPHOS gene group presented a lower mutation rate and KR/KC ratio than the mtDNA-encoded OXPHOS genes (second group).

### 2.5. Probe Panel Development and Examination

To identify the OXPHOS factors involved in mtDNA–nDNA incompatibility, we analyzed the mtDNA-encoded gene sequences and identified 10 genes with nonconserved intraspecific substitutions (Appendix A). Since the mtDNA-encoding genes are fully linked to form mtDNA haplotypes, we focused on *ND2* and *ND4* protein variants representing the haplotype variation between the *On* and *Oa* species and excluding intraspecific variants. In addition, we further examined 16 nDNA-encoded OXPHOS genes with nonconserved substitutions to detect those that represent intraspecific mutations (Figure 1 and Appendix A). We identified such species-specific polymorphisms in 11 nDNA-encoded OXPHOS genes: *ATP5C1*, *COX4B*, *COX6A2*, *NDUFA2*, *NDUFB4*, *NDUFC2*, *NDUFV3*, *UQCR4*, *UQCRB*, *UQCRC2* and *UQCRFS1*. Finally, for high-throughput screening of *On* and *Oa* samples, we designed a panel of 13 sequence probes (spanning lengths of 15–19 aa) targeting the two mtDNA and 11 polymorphic nDNA-encoded proteins (Figure 1).

Using taxonomy annotation, we screened the NCBI Short Read Archive (SRA) for *On* and *Oa* libraries, including DNA- and RNA-seq data. We identified 3834 *On* samples from 103 projects and 212 *Oa* samples from 13 projects. For these projects, we selected only those with libraries prepared from single individuals and excluded libraries of pooled samples. Additional projects were excluded in cases of low sequence coverage or quality, in which more than 5% of the probes had no hit. Other libraries were rejected because of a contradiction between the reported tilapia species and the detected mitochondrial type (probes *ND2* and *ND4*, Figure 1 and Appendix A). To avoid overrepresenting individuals of specific populations, we sampled no more than three libraries from each project or a maximum of six libraries from different projects of the same research institution. With these considerations, we retained 136 *On* and 26 *Oa* experiments from 57 and 10 projects, respectively (see Appendix A for these 67 BioProject accession numbers). These samples included SRA submissions from 49 different institutions from 16 countries (Appendix A). We used TBLASTN to compare the designed 13 probes with the selected libraries’ data under the following uniform algorithm: expect threshold 0.05, word size 5, maximum mismatches in a query 0, matrix BLOSUM62 with gap costs of existence 11 and extension 1.

Through TBLASTN screening of 136 *On* and 26 *Oa*-selected SRAs with a 13-pair panel of species-specific probes (Figure 1), we identified 64 cases in which one or both alleles of the nDNA-encoded protein originated from *Oreochromis* species which differed from the mtDNA-encoded protein it was affiliated with. This mismatch was defined as the presence of an alien allele. Thus, in the 162 SRAs analyzed experiments, the percentage of DNA-encoded alien alleles was 39.5%. Using this approach for the 162 SRA libraries, we counted 172 alien alleles for the 11 nDNA-encoded genes (an average of 15.6 ± 9.3 mismatches per gene, with alien allele frequencies ranging between 0.006 and 0.090) (Figure 1 and Appendix A). This 14.3-fold difference between the *NDUFB4* and *UQCRC2* genes exemplified this wide frequency range. A chi-square test revealed significant differences (*p* < 9.3 × 10^−9^) in the 11 values for the alien allele frequency at different loci (right column, Figure 1b). An ANOM test (analysis of means for proportions) divided the values into two subgroups (*p* < 0.05), in which the first subgroup included low values that deviated from the average of the alien allele frequencies for the three genes, *UQCRC2*, *ATP5C1* and *COX4B*. The second subgroup contained the remaining eight genes’ values that did not deviate from the average. Unpaired t-tests confirmed these subgroup differences (*p* < 4.1 × 10^−5^). A low frequency of alien alleles was detected for three nDNA-encoded genes, *UQCRC2*, *ATP5C1*, and *COX4B*, which belong to mitochondrial OXPHOS complexes III, IV, and V, respectively (Figure 1).

Alien alleles were detected in 50 of the 67 (74.6%) examined projects. Different patterns of alien alleles between the projects supported frequent and independent cases of admixture in different laboratories and commercial stocks. Cases in which the alien allele was homozygous (alien allele fixation) were detected in only three samples from the PRJNA802819 project for the genes *UQCR4* and *UQCRB* (Appendix A). Thus, assuming that selection over time favors the homozygous state of OXPHOS factors as evidence of the prevalence of homozygosity in the purebred species, the absence of mismatched allele fixation indicated that the most frequently observed admixture was recent.

### 2.6. Three-Dimensional (3D) Visualization of Mutations Associated with Negative Selection

The *UQCRC2*, *ATP5C1*, and *COX4B* variations were subunits of respiratory complexes (III, IV, and V, respectively) and were associated with negative selection that might be caused by cytonuclear incompatibility with the mtDNA-encoded subunits in these complexes (*CYTB*, *COX1*, *COX3* and ATP6, respectively), for which we detected nonconserved amino acid substitutions between *On* and *Oa*. To better analyze how these variations are oriented, we produced a 3D visualization of their relevant respiratory complexes. This was performed using the Structure tool (RCSB PDB, Mol* 3D Viewer, version 4.11.0) and the 3D structure data of assembled respiratory protein complexes deposited for *Bos taurus* (PDBs 1SQB, 5B1A, 7AJB, respectively, Figure 2). In these putative 3D models, we depicted the four nDNA-encoded non-conserved amino acid variations (marked 3, 4, 7, and 8 in Figure 2) and the expected sites of the mtDNA-encoded variations (marked 1, 2, 6, 9, and 10 in Figure 2). As shown in Figure 2, the involved subunits’ mutations are not directly interacting. Therefore, assuming that the basic 3D structure of the respiratory complexes is conserved across fish and mammals, cytonuclear incompatibility is not likely to arise from direct binding among these subunits’ mutations.

## 3. Discussion

### 3.1. The KR/KC Ratios of Oa and On OXPHOS Genes

We calculated high KR/KC ratios for the nDNA- (1.08) and mtDNA- (1.29) encoded gene groups, which indicated strong specialization of the *Oa* and *On* OXPHOS genes under adaptive selection (Appendix A). The KR/KC ratios and substitution frequencies were greater for mtDNA-encoded genes than for nuclear genes, which explains the leading role of mitochondria in the process of specialization and adaptation [49,50,51]. This is additionally supported by the high ratios of conserved and nonconserved substitutions between mtDNA- and nDNA-encoded OXPHOS genes (7 and 8.4 of conserved and nonconserved substitutions, respectively; Appendix A). Moreover, in the present work, we found that except *COX1*, all *On* and *Oa* mtDNA-encoded OXPHOS genes presented species-specific amino acid substitutions, whereas 64 proteins of 93 nDNA-encoded genes were completely identical (Appendix A). Moreover, of the 29 polymorphic nDNA-encoded genes, nonconserved substitutions were found in only 16 genes, demonstrating a KR/KC ratio of (16/29, 55.2%), which is lower than that of the mtDNA-encoded OXPHOS genes (10/12, 83.3%). This ratio variability may stem from the differences in the mutation rate attributed to the different genome properties [52]. In addition, a higher substitution rate in nDNA-encoded OXPHOS genes has been suggested to be derived from adaptive selection owing to physical interaction or proximity to the rapidly evolving mtDNA-encoded OXPHOS subunits [53,54]. Thus, only a subset of nDNA-encoded genes are subjected to physical proximity and, therefore, to positive selection.

### 3.2. Identification of nDNA-Encoded Gene Variants Associated with Hybrid Incompatibilities

In captivity, *On* and *Oa* display similar survival rates [55]. Therefore, the negative selection we observed for three nDNA-encoded factors in hybrids is not likely to be explained by selection that is specific to one of the species. Genetic incompatibility between closely related species is a major step toward complete speciation. Hybrid incompatibility arises from genetic variants that accumulate independently in diverging species populations, and when variants are brought together in hybrids, they interact ineffectively. This frequently occurs in complex functional systems, such as OXPHOS, for which several genes involved in hybrid incompatibilities have been detected [12,56]. We detected species-specific nonconserved substitutions in 11 nDNA-encoded OXPHOS genes and investigated these genes for selection trends by the frequency analysis of the alien alleles in *On* and *Oa* SRA libraries. This analysis detected a low frequency of alien alleles in genes *UQCRC2* (complex III), *ATP5C1* (Complex V) and *COX4B* (complex IV), which may indicate purifying selection against such alleles in *On* × *Oa* hybrids. The three nuclear factors (*UQCRC2*, *ATP5C1* and *COX4B*) that we identified have also been previously shown to play a role in cytonuclear incompatibility in other vertebrates. Out of the five OXPHOS protein complexes, three were involved in hybrid selection, of which complex IV has often been implicated in lethal cytonuclear incompatibility [5,57,58]. Similar to our findings, it has been shown that a single amino acid substitution in the nDNA-encoded cytochrome c (CYC) that is oxidized by complex IV is sufficient to cause hybrid breakdown [27]. Different methods and experimental designs have been used to detect genetic lethal effects in fish [12,56,59,60,61]. The latter reported embryonic mortality associated with specific combinations of mitochondria and three OXPHOS genes of complex I in *Xiphophorus birchmanni* and *X. malinchemediated* hybrids [12]. In the present study, we observed no alien allele fixation in most of the examined projects. This suggests that most hybridization events were recent. However, the fixation of *UQCR4* and *UQCRB Oa* alleles in *On* from Lake Hora (Ethiopia, PRJNA802819) [62] may indicate ancient admixing in this lake (samples SRX14028738-40, Appendix A).

### 3.3. Mitochondrion Complex IV

The *COX4* autosomal gene encodes the largest subunit of OXPHOS complex IV, which physically interacts with the mtDNA-encoded subunit *COX2*, at numerous sites [63]. In tilapia nDNA, the *COX4* gene has two paralogs, *COX4A* and *COX4B,* which encode moderately similar (51% identity) polypeptides of 174 and 169 amino acids, respectively. We observed a single nonconserved amino acid substitution in the *COX4B* protein *On* and *Oa* orthologs, for which the allele frequency analysis indicated a cytonuclear incompatibility, whereas the *COX4A* protein paralogs were fully conserved. Having evolved to encode proteins with slim similarity (<55%), paralogous gene duplications and even triplications of OXPHOS genes are more common in teleosts than in mammals. This phenomenon raises two important questions: how can such remarkably divergent paralogs interact with the same set of mtDNA-encoded factors, and how does this interaction influence selection in hybrids [64]? It has been suggested that duplication of OXPHOS genes is followed by specialization of paralogs for different functions in different tissues [65] at different developmental stages [66]. Thus, understanding the selection process that drives hybrid survival should involve identifying the tissues in which *COX4B* is expressed over *COX4A*, as the loss of function of one paralog may be partially compensated by another paralog [57], attenuating the negative selection induced by the cytonuclear incompatibility. Notably, different environments induce different ratios of the *COX4* paralogs by modulating the expression of Lon proteases, which preferentially degrade *COX4A* [67].

In mammals, *COX4* has been more strongly implicated in cytonuclear incompatibility than other nuclear *COX* genes [57,58]. However, some differences exist between high- and low-vertebrate species. Mammalian *COX4A* expression is mainly constitutive, expressed across most tissues, whereas *COX4B* presents more of a tissue-specific expression pattern. The promoter elements that confer oxygen sensitivity in mammalian *COX4B* are highly conserved but not in lower vertebrates. In fish, *COX4B* promoters are not hypoxia-responsive [63]. However, an interesting pattern of subfunctionalization of *COX4* paralogs in *On* has been proposed following analysis of the *COX4A*/*COX4B* expression ratio in the brain, heart, liver, red muscle and kidney under low (3%) and high (21%) oxygen levels. In *On* fish, which prefer an oxygen-rich environment (Figure 1a), hypoxia lowers the expression ratio [68]. Thus, hybrids that inherit the *Oa* preference for bottom-dwelling and the *On COX4B* allele would be selected against, owing to cytonuclear incompatibility.

To determine nonconserved substitutions, we used classic amino acid classification, which is based on their structure and the general chemical characteristics of their side chains [25,26]. However, to determine the best association between amino acid properties and selective pressure, alternative amino acid classifications have been proposed by different studies [23,24]. However, for the three genes (*UQCRC2*, *ATP5C1* and *COX4B*), the classical classification was the most sensitive.

### 3.4. Mitochondrion Complex V

Similar to *COX4B*, we detected negative selection against *ATP5C1* alien alleles. Among the 20 nDNA-encoded mitochondrial ortholog ATPases, seven nonconserved substitutions were detected between *On* and *Oa*, in four genes, *ATP5C1*, *ATP5D*, *ATP5I* and *ATPIF* (Appendix A). Interestingly, in these 20 ATPases, we observed only 6 conserved substitutions, indicating a high ratio of nonconserved/conserved substitutions (7/6 = 1.17). This preference for functional variation suggests positive selection mediated by the benefits of specialization and adaptation of mitochondrial ATPases to different environments. Each of these nonconserved substitutions can potentially lead to cytonuclear conflict in *On* × *Oa* hybrids. Hybrid selection in crosses of two close species of Atlantic eels (*Anguilla anguilla* and *A. rostrate*) has been reported to be mediated by the compatibility of mtDNA- and nDNA-encoded factors for ATPases of mitochondrial complex V, especially *ATP5C1* [69]. In these hybrids, negative selection was most evident for the combination of the nDNA-encoded *ATP5C1* and the mtDNA-encoded *ATP6*, likely due to physical incompatibility between these proteins, possibly creating a barrier between these two Atlantic eel species, which naturally hybridize [69,70]. A similar phenomenon has been reported for two subpopulations of a marine bivalve (*Limecola balthica*), including sequence conservation among *ATP*ase genes, a high ratio of nonsynonymous/synonymous substitutions, and incompatibility of *ATP5C1* [71].

### 3.5. Mitochondrion Complex III

In the present work, the nDNA-encoded OXPHOS complex III subunit, *UQCRC2,* showed the strongest purifying selection for the *On*/*Oa* hybrids. We also detected additional nonsynonymous amino acid substations in the three other complex III nDNA-encoded genes (*UQCR4*, *UQCRB*, and *UQCRFS1*). However, in contrast to *UQCRC2*, we did not observe a signal of negative selection associated with these genes in hybrids. Under hypoxia or glucose deprivation, the coregulation of complexes III and IV is lost. This finding was demonstrated by analyzing changes in the neuronal proteome in oxygen- and glucose-deprived SH-SY5Y cell cultures, which revealed significant downregulation of *UQCRC1* and *UQCRC2* expression but upregulation of *ATP5* and *ATP6* expression [72]. It has been suggested that a decrease in *UQCRC1* and *UQCRC2* expression under chronic stress conditions allows cells to reduce reactive oxygen species [73]. Moreover, following dysregulation of OXPHOS, some mitochondrial proteins are redirected to lipid and ketone metabolism as alternative sources of energy [72]. Thus, coordinated *UQCRC2* and *ATP5C* regulation may be essential for OXPHOS function in hybrids.

### 3.6. Cytonuclear Incompatibility Examination by 3D Visualization

To visualize the mutations associated with negative selection, we chose cattle models because advanced 3D modeling for respiratory III–IV complexes and supercomplexes is only available for a few vertebrate organisms, all of which are *Artiodactyla* mammals [74]. Using this modeling, we demonstrated that the involved subunits’ mutations are far apart and thus are not likely to display interference in non-covalent residue–residue contacts that would mediate cytonuclear incompatibility. The most striking example of this is the mutation A17V in *COX4B*, which is depicted on the mitochondrial transit peptide of *btCOX4I1* (Figure 2), which being cleaved off should entirely not interact with complex IV assembly. However, this presentation (Figure 2) is over-simplified, and it does not account for different state conformations, transient reactions, and interactions within the super complex. Indeed, studies have demonstrated that cytonuclear incompatibility is mediated by nDNA-encoded OXPHOS proteins that do not directly interact with mtDNA-encoded subunits and even by genes upstream of the OXPHOS genes [75,76]. Therefore, cytonuclear incompatibility may arise from complex interactions, which our current knowledge cannot predict.

### 3.7. Further Research

Identifying signals of negative selection on alien *UQCRC2*, *ATP5C1*, and *COX4B* gene alleles and developing a panel of species-specific probes presents an opportunity for further experimentation. In hybrids, in addition to incompatibility between major OXPHOS proteins, the low efficiency of the energy production process may result from incompatibility with other factors such as hormones and transcription factors [77]. Analysis of data of *Oa* × *On* F2 crosses via genome-wide association studies (GWASs) may further identify gene–allele combinations of these factors that would be suitable for coping with different stress conditions.

## 4. Materials and Methods

### 4.1. Identifying the OXPHOS Genes in the On and Oa Mitochondrial Genomes

Tilapia sequences of mitochondrial *COX1* reference fragments determined in a previous study [46] were used to BLAST search for *On* and *Oa* nucleotide submissions. To detect misidentification of *Oreochromis* species in the NCBI and BOLD databases, we analyzed 14 complete mitochondrial sequences reported as *On* and *Oa* in the NCBI, and these sequences were used to compare the sequences of *On* and *Oa* for 13 mDNA-encoded OXPHOS proteins (Appendix A). Only complete mitochondrial sequence reports that matched the species reference of the *COX1* sequence were analyzed, whereas mismatching reports were removed from further analysis.

### 4.2. Identifying the nDNA-Encoded OXPHOS Genes in On and Oa

NDNA-encoded OXPHOS genes were detected in the NCBI *On* (nucleotide accession GCF_001858045.2) and *Oa* (nucleotide accession GCF_013358895.1) genomes by text searching with symbol names of the OXPHOS mammalian orthologous genes, which have been described previously [7]. The protein sequences of the detected tilapia genes were used again in a TBLAST search against the genome builds of *On* and *Oa* to detect possible additional copies of the OXPHOS genes and to obtain proper gene annotations for these paralogs.

### 4.3. Orthologous Genome Positions of Nuclear Genes

The existence of paralogous OXPHOS genes might hamper the identification of true orthologous genes. To ensure true identification of the *Oreochromis* OXPHOS genes, we compared the mapping positions of nuclear genes in both the *On* and *Oa* genome builds. For genes whose positions were assigned to unmapped scaffolds in the *Oa* genome, we examined the *On* mapping positions of other genes in these scaffolds under the assumption that unmapped genes are located near their syntenic paralogs in the *On* genome.

### 4.4. Identifying Nonsynonymous Substitutions in OXPHOS Genes

Using Needleman–Wunsch alignment of paired protein sequences of *On* and *Oa* 93 nuclear and 13 mitochondrial orthologous genes, which encode OXPHOS factors, we scored conserved and nonconserved amino acid substitutions. The amino acids were classified into six classes: aliphatic (G, A, V, L, I); hydroxyl or sulfur/selenium-containing (S, C, U, T, M); cyclic (P); aromatic (F, Y, W); basic (H, K, R); and acidic and amide-containing (D, E, N, Q) [25,26]. The frequencies of these substitutions were recorded (Appendix A) and compared between nuclear and mitochondrial OXPHOS genes. We also divided the number of amino acid substitutions by the length of the protein sequences to obtain the relative rates of these substitutions (Appendix A).

### 4.5. Probe Panel Development and Examination

All 70 nonconserved amino acid substitutions in the 16 nDNA-encoded and 10 mtDNA-encoded OXPHOS proteins were initially used to develop a panel of species-specific *On*/*Oa* probes. For these proteins, on the basis of their amino acid sequences, we designed 3 probes of 19 residues, including the middle, left, and right terminal positions of the variable amino acid, respectively. These initial probes were used as queries for the TBLASTN search for five *On* (ERX6793970, ERX7626946, DRX094401, SRX14028738 and ERX2765256) and five *Oa* (ERX2240356, SRX8298259, SRX7899546, SRX20735086 and SRX19920617) SRA experiments. Considering that mitochondrial amino acid substitutions are tightly linked, we selected only two pairs of probes for mitochondria (for the ND2 and ND4 proteins, Figure 1 and Appendix A). Among the 16 nuclear genes, 11 genes (*ATP5C1*, *COX4B*, *COX6A2*, *NDUFA2*, *NDUFB4*, *NDUFC2*, *NDUFV3*, *UQCR4*, *UQCRB*, *UQCRC2*, and *UQCRFS1*) were successfully designed as species-specific probes. Probes in the remaining five genes with nonconserved substitutions likely presented intraspecific variability rather than interspecific variability and were therefore irrelevant for further analysis. Thus, a panel of 13 pairs of probes (2 mitochondrial and 11 nuclear) were prepared after additional optimization of the probe length by choosing the length that generates the maximal interspecific discrimination results in a TBLAST search (Figure 1 and Appendix A). JMP6 statistical software (release 6.0.0, SAS Institute Inc., Cary, NC, USA) was used for chi-square tests to analyze the 11 frequency values of the alien alleles (Figure 1b, right column). To analyze the statistical distribution of these data, the first test was used to detect deviations from the normal frequency distribution in *On* or *Oa* and in their combined data. Following the identification of subgroups associated with different alien allele frequencies via the first test, the second ANOM test was performed to sort these frequencies into subgroups. The third test was carried out to determine the significance of the difference between the detected subgroups.

### 4.6. Three-Dimensional (3D) Visualization

Bovine protein subunits and amino acid positions orthologous to those of tilapia were identified using NCBI’s protein BLAST tools [78]. Mol* 3D Viewer [79] was used to visualize PDB models of respiratory III–IV complexes (1SQB, 5B1A, 7AJB) using the defaults and the following options: the components were polymers with added molecular surface representation; water, ligands, etc. were set to hidden; and the Chain property under set coloring was assigned to Chain ID. To highlight mutations, the mouse pointer was used to select the mutation in the polypeptide sequence of the selected subunit entity and chain instance. Photos of highlighted regions were superimposed using Adobe Photoshop cs2 (Adobe Systems, San Jose, CA, USA).

## 5. Conclusions

Identifying negative selection against OXPHOS alien alleles in the *UQCRC2*, *ATP5C1* and *COX4B* genes via our panel of species-specific probes provides an opportunity for further experimentation with populations of *On* × *Oa* F2 hybrids. Monitored under different stress conditions, the resulting allele segregation of these gene orthologs would enhance the understanding of the critical combination of OXPHOS genes for each stress condition. This knowledge could aid in the rapid development of commercial hybrid populations adapted to specific climatic and rearing conditions.

## Figures and Tables

**Figure 1 ijms-26-02089-f001:**
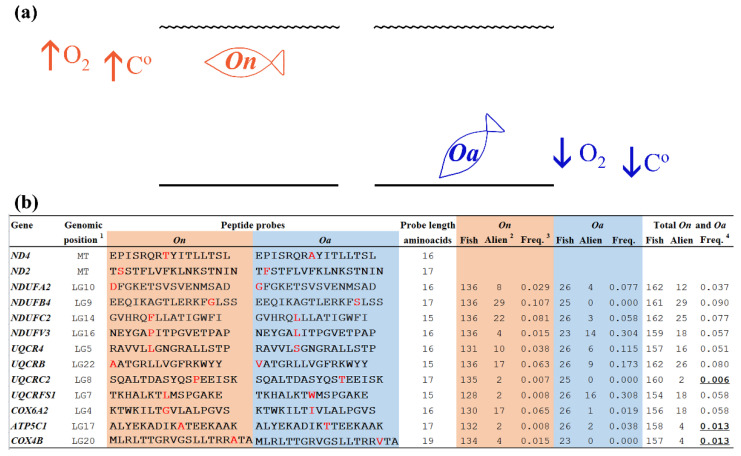
Variation between *O. niloticus* (*On*) and *O. aureus* (*Oa*). Two *Oreochromis* species important for the aquaculture industry were compared: (**a**) Schematic representation of the key differences between *On* and *Oa*, which are associated with a preference for a specific water depth, adaptation to different oxygen concentrations and temperatures, and feeding habits, with *On* and *Oa* having preferences for omnivorous and herbivorous diets, respectively; (**b**) nDNA-encoded OXPHOS genes with nonconserved variations (red font) between the *On* and *Oa* ortholog genes were used to determine peptide probes. These allele probes were applied for TBLASTN searches against 162 SRA libraries annotated as *On* and *Oa* to determine their frequencies. The mtDNA-encoded probes were used to verify the species annotation. The table shows the basic statistics of these analyses, ^1^ Genomic position in *On* genome build (nucleotide accession GCF_013358895.1). ^2^ Number of cases in which one or two foreign alleles were detected. ^3^ Frequencies of alien alleles among all the fish alleles. ^4^ A bold underlined font denotes a significantly (*p* < 10^−4^) lower frequency than expected. Three genes that presented a low frequency of alien alleles are denoted by bold font in the last column.

**Figure 2 ijms-26-02089-f002:**
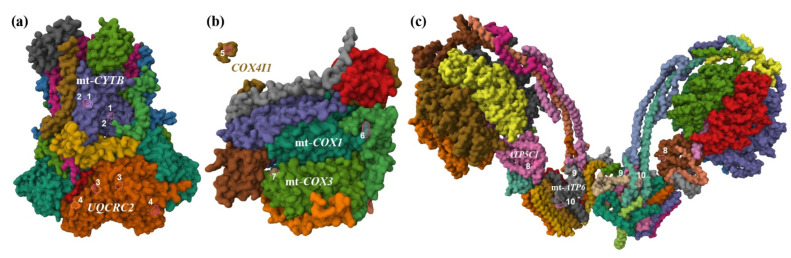
Diagram deciphering position of 13 non-conserved amino acid variations between *O. niloticus* (*On*) and *O. aureus* (*Oa*). Three-dimensional (3D) visualization of these variations was performed using Mol* 3D Viewer (RCSB PDB, Mol* 3D Viewer, version 4.11.0) and the 3D structure data of assembled respiratory protein complexes, which were deposited for *Bos taurus* in the protein data bank format (PDB). The protein complexes are displayed following their orientation on the inner mitochondrial membrane and colored by the Chain Id default option. The relevant gene symbols of the variable proteins are shown, and variations (1–10) are indicated next to the orthologous residues highlighted by the software tool. (**a**) Complex III (bovine PDB 1SQB). Each polypeptide within the complex has two instances (colored similarly) and these are as follows (from top left to bottom right): subunit *UQCRH* (grey,); *UQCRQ* (light brown); mtDNA-encoded cyc1 *UQCR4* (bordeaux); *UQCRFS1* (green I); *UQCR10* (blue); mtDNA-encoded subunit *CYTB* (purple); *UQCR11* (green II); *UQCRB* (yellow); *UQCRC1* (green III); proteolytic procced polypeptide of *UQCRFS1* (red); *UQCRC2* (orange). With two instances each, four variable amino acids are denoted as follows (mutation positions are of the *On* and *Oa* orthologous protein, GenBank protein accessions: ADD71229, ADD71255, and XP_003438780, XP_031595481, respectively): S360F (1), F365L (2); T258A (3); P404T (4). (**b**) Complex IV (bovine PDB 5B1A). *COX4I1* (dark yellow); *COX6C* (grey); *COX5A* (red); mtDNA-encoded *COX2* (purple); mtDNA-encoded *COX1* (green III); *COX5B* (green II); *COX6B* (brown); mtDNA-encoded *COX3* (green I); *COX6A2* (orange); *COX7A1* (light brown); completely hidden behind are *COX7B*, *COX7C*, *COX8B* (pink, yellow, and grey, respectively). Three variable amino acids are denoted as follows (mutation positions are of the *On* orthologous protein GenBank protein accessions: XP_003457010, XP_031584659, ADD71219, ADD71232, ADD71171, and ADD71236, respectively): mutation on an illustration of the cleaved mitochondrial transit-peptide A17V (5), A516T, G521E and T524A are three mutations on the c-terminal, which is outside of the homologous region with cattle; thus, the c-terminal end is indicated (6); A107V (7). (**c**) Complex V (bovine PDB 7AJB). Each polypeptide within the complex has at least two instances (colored differently), and these are as follows (from top left to bottom right): *ATP5O* (brown, yellow); *ATP5F1* (light brown, turquoise); *ATP5J* (bordeaux, purple); *ATP5A1L* with 6 instances (dark yellow, green II, light brown, bordeaux, grey, green I); *ATP5B* with 6 instances (purple, red, orange, blue, yellow, green II); *ATP5H* (pink, light blue), *ATP5C1* (pink, brown); *ATPIF* (green I, pink); *ATP5E* (light blue, light brown); *ATP5D* (turquoise, grey); *ATPG2* with 16 instances (yellow, dark yellow, beige, light brown, grey, grey II, green III, red, orange, blue, purple, green II, bordeaux, purple II, green I, orange); mtDNA-encoded *ATP6* (grey, pink); mtDNA-encoded *ATP8* (orange, green III); *ATPMJ* (dark yellow, green I); *ATP5MG* (orange, green II); *ATP5ME* (yellow, light green); *ATP5MK* (grey, light brown). With two instances each, four variable amino acids are denoted as follows (mutation positions are of the *On* and *Oa* orthologous protein, GenBank protein accessions: XP_003448120, XP_031602241, ADD71138, and ADD71248, respectively): A79T (8), A136T (9), adjacent A136T and P137L (10).

**Table 1 ijms-26-02089-t001:** Amino acid variation between *O. niloticus* and *O. aureus* OXPHOS proteins.

Encoding Type	Variable/ Total Proteins ^1^	% Variable Proteins	Variability ^2^ ± SD	*p* ^3^
Nuclear	16/93	17.2	0.16 ± 0.51	0.01
Mitochondrial	10/13	76.9	1.08 ± 1.09	

^1^ Variable protein is denoted when the orthologs display nonconserved substitutions. ^2^ Mean rate of nonconserved amino acid substitutions in protein types. ^3^ The *t*-test probability for the difference between means of nonconserved amino acid substitution rates between mtDNA- and nDNA-encoded proteins.

## Data Availability

The data presented in this study are available in GenBank at https://www.ncbi.nlm.nih.gov (accessed on 24 February 2025). These data were derived from the following resources available in the public domain: https://www.ncbi.nlm.nih.gov/gene/; https://www.ncbi.nlm.nih.gov/sra/ (accessed on 24 February 2025); and http://www.ensembl.org/index.html (accessed on 24 February 2025).

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
