# Peer review of "Negative Selection in Oreochromis niloticus × O. aureus Hybrids Indicates Incompatible Oxidative Phosphorylation (OXPHOS) Proteins"

_ijms, 2025, doi:10.3390/ijms26052089_

Round 1
Reviewer 1 Report
Comments and Suggestions for Authors
This paper examined the mutations in OXPHOS in the hybrids of two close tilapia species based on the currently available SRA data, aiming to identify the potential mito-nuclear incompatibility. The results and analyses of this study will facilitate our understanding of cytonuclear incompatibility, negative selection, and reproductive barriers caused by the malfunction of the energy production system in interspecific hybrids. The study was well designed and performed and the paper was well written. I would recommend publishing with minor revisions. My specific comments are listed below.
1. I understand that the authors identify the species of a certain SRA library by its mt-DNA, if there is a mutation in the nDNA, how can the authors determine whether it comes from hybridization or it is just an intraspecific mutation?
2. How to define cytonuclear incompatibility? This should be defined in Intro.
3. Some key data should be given in more detail. For example, the authors stated ‘We observed a single nonconserved aminoacid substitution in the COX4B protein On and Oa orthologs, for which the allele frequency analysis indicated a cytonuclear incompatibility'. The authors should provide a diagram to decipher the mutations and explain how this caused cytonuclear incompatibility.
4. Aminoacids should be changed to amino acids.
5. I suggest that the authors compare the major traits of the two species in a table instead of a simple figure in Fig. 1a.
his caused cytonuclear incompatibility.
Reviewer 2 Report
Comments and Suggestions for Authors
- This paper entitled: “Incompatible oxidative phosphorylation (OXPHOS) proteins mediate negative selection in Oreochromis niloticus × aureus hybrids” is interesting in its subject matter, but there are major issues. The major problem is that the overall conclusion is not supported by the data presented. The work really presented no data on incompatibility of any kind, but just sequence analysis of genes involved in oxidative phosphorylation. Using KR/KC and Ka/Ks analysis, the authors can reason that negative selection may be involved but provide no clue about hybrid incompatibility. Therefore, the title needs to be re-phrased to reflect the support of the data and the conclusions they can draw. Because of this major problem, the paper really needs to be re-written such that the work is presented as a sequence analysis paper.
- The authors claimed that the sequence identities are wrong from NCBI and other sources. This needs to be carefully addressed. The best to coordinate a community to eliminate or correct the sequences or their identities. Without clear evidence, it is dangerous to just claim the work done by others are wrong.
- Abstract: “Hybrid tilapias escaping to the wild are likely to impact natural populations. Hybrids competing with wild populations undergo selection for different stressors, e.g., oxygen levels, salinity, and low-temperature tolerance.” This is so isolated that it should not be included. In no places the authors conducted any analysis of escapes and its impact on natural stocks, although this might be true. Therefore, this needs to be deleted.
- The results were presented as if they are materials and methods. The authors need to delineate methods, materials, and results that include the findings of the analysis.
Line 159, aminoacid should be amino acid
Line 279, aminoacid should be amino acid
Author Response
Please, find attached.

Round 2
Reviewer 2 Report
Comments and Suggestions for Authors
My concerns have been addressed